# Unveiling ctDNA Response: Immune Checkpoint Blockade Therapy in a Patient with POLE Mutation-Associated Early-Onset Colon Cancer

**DOI:** 10.3390/curroncol32070370

**Published:** 2025-06-25

**Authors:** Ramya Ramachandran, Marisa Cannon, Supriya Peshin, Madappa Kundranda, Aaron J. Scott

**Affiliations:** 1Department of Internal Medicine, University of Arizona, Tucson, AZ 85724, USA; ramya.ramachandran2@bannerhealth.com; 2College of Medicine, University of Arizona, Tucson, AZ 85724, USA; mecannon@arizona.edu; 3Department of Internal Medicine, Norton Community Hospital, Norton, VA 24273, USA; supriyapeshin720@gmail.com; 4Division of Cancer Medicine, Banner MD Anderson Cancer Center, Gilbert, AZ 85234, USA; madappa.kundranda@bannerhealth.com; 5Division of Hematology/Oncology, University of Arizona Cancer Center, Tucson, AZ 85724, USA

**Keywords:** POLE, early onset colorectal cancer, Nivolumab, Ipilimumab, ctDNA, case report

## Abstract

Colorectal cancer is one of the most common cancers, and alarmingly, an increasing number young people under 50 are being diagnosed with it. The reasons for this are not fully understood, but certain genetic changes may play a part. One of these involves a gene called POLE, which normally helps to repair DNA mistakes. When POLE is mutated, cells collect too many errors, but this also makes the cancer more visible to the body’s immune system. This report details a case of a young man with advanced colorectal cancer carrying the POLE mutation who did not respond to regular chemotherapy. However, once he started immunotherapy, a treatment that helps the immune system to fight cancer, his tumors shrank dramatically. Blood tests showed that his cancer-related DNA nearly vanished, and two years later, he remains cancer-free. This story shows how understanding a tumor’s unique genetics can lead to better, more personalized treatments, and offers hope for patients facing similar situations in the future.

## 1. Introduction

The anticipated surge in early-onset colorectal cancer (EOCRC) rates is alarming, with a projected 90% increase in individuals aged 20–24 and 27.7% increase for those aged 35–49 by 2030 [1]. EOCRC, typically observed in those under 50, poses unique challenges, with limited knowledge of its risk factors, despite some known factors including Caucasian ethnicity, family CRC history, hyperlipidemia, and obesity [2]. While hereditary factors such as Familial Adenomatous Polyposis (FAP) and Lynch syndrome are linked to EOCRC, three out of four individuals affected have no familial history and their cancer is classified as sporadic. EOCRC is associated with distinct clinicopathologic characteristics, including being mucinous, being poorly differentiated, having signet ring histology, and having tendencies toward high-grade tumors and lymphatic invasion, as noted by Chang et al. [3]. Moreover, EOCRC is characterized by a more distal tumor location and a later TNM stage at presentation compared to late-onset CRC (LOCRC). Some ongoing studies are focusing on highly expressed genes like secreted frizzled-related protein 4 (SFRP4) and cartilage oligomeric matrix protein (COMP), both of which have been shown to play a role in cancer invasion and metastasis [1]. Notably, the gut microbiome’s role in EOCRC remains underexplored, highlighting a vital area for future research [1]. DNA Polymerase δ and ε (POLE) are both high-fidelity enzymes that play critical roles in the formation of new DNA strands, the regulation of these strands during the cell cycle, and the preservation of existing DNA strands [4]. They play a major role in leading-strand synthesis, have 3′🡪5′ exonuclease activity, recognize and repair mismatched bases through proofreading activity via their exonuclease region, and participate in nucleotide excision repair and double-strand break repair [4]. POLE mutations have been associated with various other genetic mutations, such as phosphatase and tensin homolog (PTEN) mutations and dMMR mutations [4]. The specific combination of genes associated with POLE mutations can impact the type of cancer that develops, the resulting diagnosis, and the appropriate treatment strategies [4]. Mutations in the exonuclease domain of DNA Polymerase ε, particularly those causing a cascade of downstream mutations, increase neoantigen production [5]. Heightened DNA mutation/megabase, also known as a high tumor mutation burden (TMB), and neoantigen production lead to increased immune-cell infiltration [5,6]. POLE-mutated and microsatellite-instability (MSI) tumors upregulate the expression of immune checkpoints such as programmed death protein-1, (PD-1), programmed death ligand-1 (PD-L1), and Cytotoxic T lymphocyte-associated protein (CTLA-4) in response to the increased infiltration of immune cells caused in part by POLE mutations [7]. This phenomenon makes them particularly susceptible to immune checkpoint inhibitors [7]. However, more clinical data is required regarding the response of POLE tumors to immune checkpoint inhibitors [5]. Our paper aims to showcase the effectiveness of nivolumab and ipilimumab in treating POLE-mutant CRC, as demonstrated by a rapid decrease in ctDNA, radiographic response, and normalization of CEA in this patient with EOCRC.

## 2. Case Report

A 45-year-old male with a past medical history of lactose intolerance presented to his primary care physician with a five-week history of right lower-quadrant abdominal pain, bloating, swelling, discomfort, and increased flatulence. On physical examination, a visible and palpable mass was noted in the right lower quadrant. A contrast-enhanced computed tomography (CT) scan of the abdomen and pelvis revealed an 8 cm mass arising from the cecum, accompanied by presumed metastatic mesenteric lymphadenopathy and potential early invasion of the right lower-quadrant anterior abdominal wall musculature (Figure 1).

Subsequent colonoscopy identified a large cecal mass. Histopathological analysis of biopsy samples revealed a moderately to poorly differentiated adenocarcinoma. Microscopy demonstrated malignant cells with irregular, hyperchromatic nuclei, prominent nucleoli, and frequent mitotic figures (Figure 2). Immunohistochemistry confirmed intact nuclear expression of all mismatch repair (MMR) proteins, indicating microsatellite-stable (MSS) status. Immunohistochemical staining for mismatch repair (MMR) proteins showed intact nuclear staining, indicating microsatellite-stable (MSS) status. PD-L1 immunohistochemistry (22C3 pharmDx assay) was also performed on the primary tumor and revealed a tumor proportion score (TPS) of <1%, with minimal immune-cell staining. Despite low PD-L1 expression, the decision to proceed with immunotherapy was supported by the presence of an ultra-mutated POLE mutation and a high TMB, both recognized as predictive biomarkers for immune checkpoint inhibitor response.

The patient was initially given neoadjuvant systemic therapy with FOLFOX (folinic acid, 5-fluorouracil, oxaliplatin). After two months of treatment, repeat CT imaging demonstrated a reduction in the size of the cecal mass and mesenteric lymph nodes, and his serum carcinoembryonic antigen (CEA) level had decreased to 2.3, indicating a favorable initial response. A laparoscopic-assisted right hemicolectomy with anastomosis and omentectomy was performed. The surgical pathology demonstrated a T4 tumor with invasion into the abdominal wall, though the resection margins were negative. However, a postoperative CT scan of the abdomen and pelvis performed three days later revealed new multiple hepatic metastases (Figure 3).

Comprehensive molecular profiling of the resected tumor was performed. The results showed a high tumor mutational burden (TMB) of 87.9 mutations/megabase, microsatellite stability, POLE mutation, BRAF p.D594G mutation, and PIK3CA p.R88Q missense mutation. Based on the presence of metastatic disease, the patient was started on first-line systemic chemotherapy with FOLFOXIRI plus Bevacizumab, in accordance with standard treatment guidelines. After three cycles, clinical and radiographic assessment revealed disease progression, evidenced by worsening abdominal pain, rising CEA levels (145.7), and an interval increase in the hepatic metastatic burden on imaging (Figure 4).

Given the tumor’s POLE mutation, high TMB, and refractory response to cytotoxic therapy, the patient was considered for immunotherapy. He was initially started on single-agent pembrolizumab, an anti-PD-1 monoclonal antibody. Although his CEA and ctDNA levels rose during the initial treatment, pseudoprogression was suspected. Due to the patient’s preference and clinical suspicion of delayed immunologic response, therapy was escalated to combination immune checkpoint blockade with nivolumab (anti–PD-1) and ipilimumab (anti–CTLA-4). After two cycles, follow-up CT demonstrated further enlargement of hepatic lesions and persistent splenomegaly (Figure 5), along with a CEA increase to 183.4.

Despite radiographic progression and rising CEA levels, Signatera™ ctDNA testing revealed a dramatic decline in tumor-derived DNA fragments, supporting the diagnosis of pseudoprogression (Figure 6). Due to the development of grade 3 immune-mediated hepatitis, ipilimumab was discontinued, and the patient was transitioned to single-agent maintenance treatment with nivolumab.

Within one month of transitioning to single-agent nivolumab, the patient’s CEA level decreased to 38.4. He continued on maintenance nivolumab for two years. He remains in serologic remission, with undetectable ctDNA and normal CEA levels. A follow-up CT of the abdomen and pelvis showed no radiologic evidence of active malignancy (Figure 7).

## 3. Discussion

Within the epidemiological landscape, POLE exonuclease domain mutations are identified in only 0.3–0.7% of colorectal cancer and polyposis cases. In a multicenter retrospective biomarker study, individuals harboring this mutation were more likely to be younger males (75.8% vs. 55.5%), had a higher prevalence of right-sided tumors (68.8% vs. 39.8%), and were more frequently diagnosed at earlier stages of disease [8]. While the POLE mutation is associated with an increased tumor burden and various other genetic deficiencies, the increased tumor immunogenicity enhances responsiveness to immune checkpoint inhibitors and offers a positive predictive biomarker for IO use. The heightened presence of CD8 and CD4 T cells, natural killer cells, and Th1 cells, along with elevated levels of CXCL9 and CXCL10, makes patients with MSI and POLE-mutated tumors prime candidates for immunotherapy in advanced stages [9,10]. A retrospective study by Mao et al. of 295 people with stage II CRC showed that patients with POLE mutations were more prone to lymphovascular invasion (55.6% vs. 17.1%, *p* = 0.003) and a higher tumor mutation burden (TMB) compared to patients who harbored POLE wild-type tumors [11]. Analysis of colorectal liver metastases with POLE mutations showed a high TMB, demonstrating response to IO [12]. This is in direct contrast to IO responses in patients with POLE wild-type colorectal liver metastases, for which response rates are unavailable. The case involved a young patient diagnosed with high-grade, metastatic colorectal cancer. While there have been prior reports of positive responses to pembrolizumab in patients with CRC with POLE mutations, the responses reported were not as complete or durable as that observed in this patient [7]. Hence, a strategic transition to using off-label dual-agent immunotherapy was conducted to enhance the immunotherapy response. The exceptional response was tracked through dramatic declines in CEA and ctDNA levels. It is debatable whether the cytotoxic effects of chemotherapy may actually contribute to the immunogenicity and thus responsiveness to IO; we do not believe that this was the case in this patient’s experience, and the combination of IO plus chemotherapy remains an area of investigation [7,13,14]. This case is one of the few demonstrating a complete and enduring response to immunotherapy in sporadic, metastatic, POLE-mutant EOCRC. Emerging evidence suggests that interactions with the activities of other polymerases, such as POLQ and REV3L, may modulate the mutational landscape and immune response in POLE-mutated tumors, and these are areas for future research [15]. While PD-L1 expression in the primary tumor was low (TPS < 1%), this aligns with recent findings that in POLE-mutated colorectal cancers, PD-L1 may not be a reliable biomarker of immunotherapy response. Instead, the elevated neoantigen load resulting from polymerase proofreading deficiency may be the dominant driver of immune activation [16,17,18,19]. Our case supports the concept that POLE mutation and high TMB can override traditional markers like PD-L1, guiding effective immunotherapy use even in the absence of high PD-L1 expression [16,18,19].

## 4. Conclusions

This case illustrates a profound and sustained response to immune checkpoint inhibition in a patient with early-onset, microsatellite-stable, POLE-mutated colorectal cancer. Despite low PD-L1 expression and initial radiographic progression, a rapid decline in ctDNA and CEA confirmed treatment efficacy and long-term remission. This highlights the critical role of POLE mutations and high TMB as predictive biomarkers for immunotherapy treatment, and supports the utility of ctDNA in guiding immunotherapy response.

## Figures and Tables

**Figure 1 curroncol-32-00370-f001:**
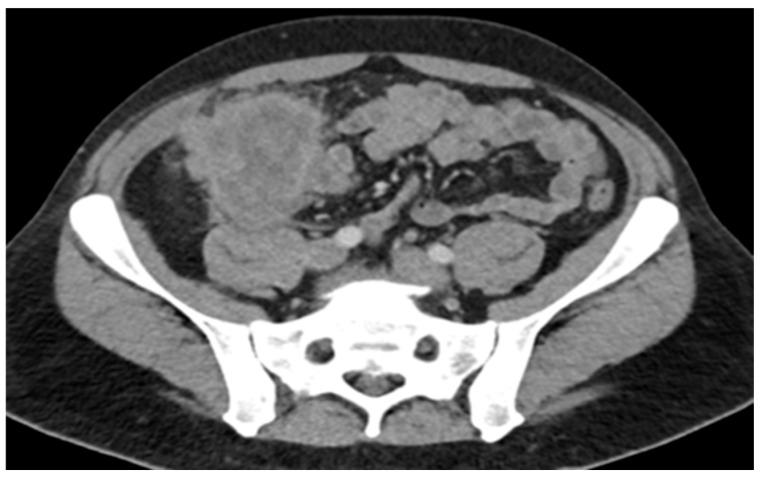
Mass with dimensions of 8 cm in cecum; presumed metastatic mesenteric lymphadenopathy with possibility of early invasion of RLQ anterior abdominal wall musculature.

**Figure 2 curroncol-32-00370-f002:**
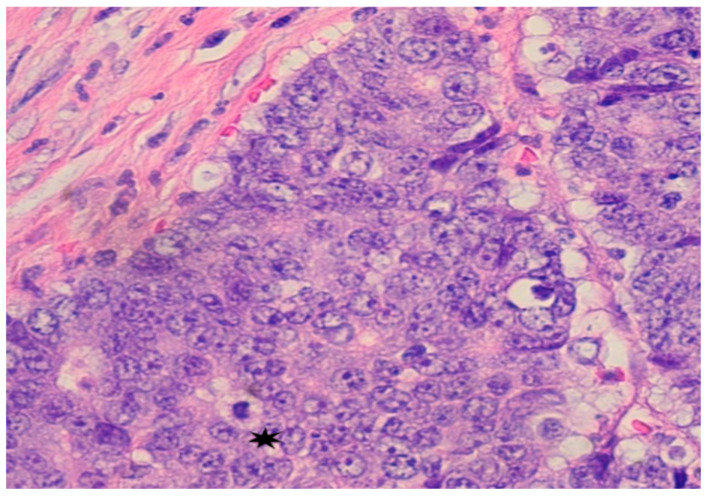
Pathology from Colonoscopy: Mass in cecum identified as poorly differentiated adenocarcinoma; malignant cells show large and irregular nuclei with prominent nuceoli and mitotic figures *.

**Figure 3 curroncol-32-00370-f003:**
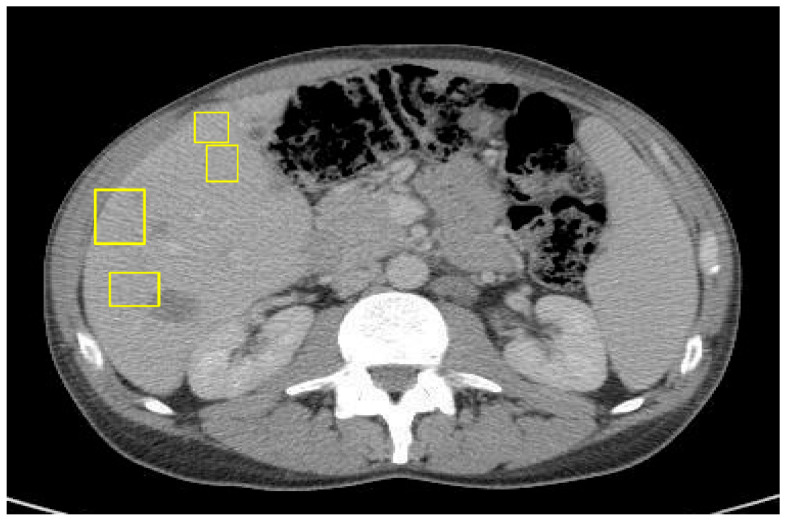
CT of abdomen/pelvis with IV contrast 3 days post right-sided hemicolectomy of primary colon cancer with curative intent.

**Figure 4 curroncol-32-00370-f004:**
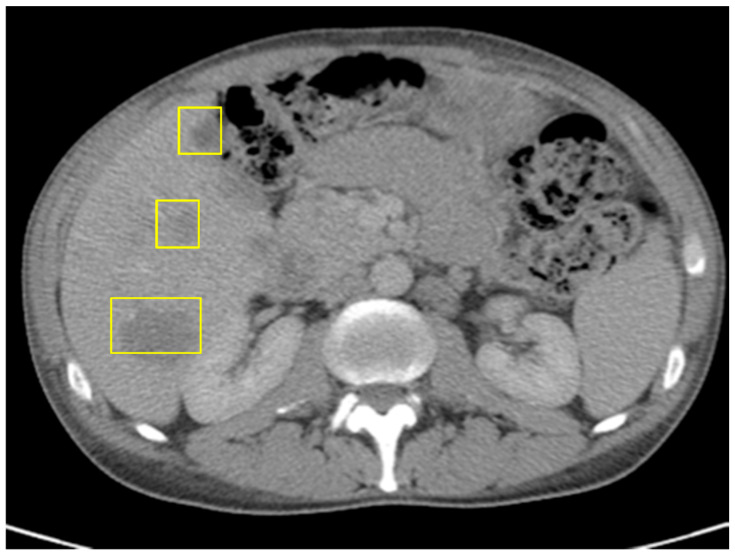
CT of abdomen/pelvis just before ICI: increased size of hepatic metastases, splenomegaly, and unchanged prominent left para-aortic lymph nodes; no evidence of metastatic disease in chest.

**Figure 5 curroncol-32-00370-f005:**
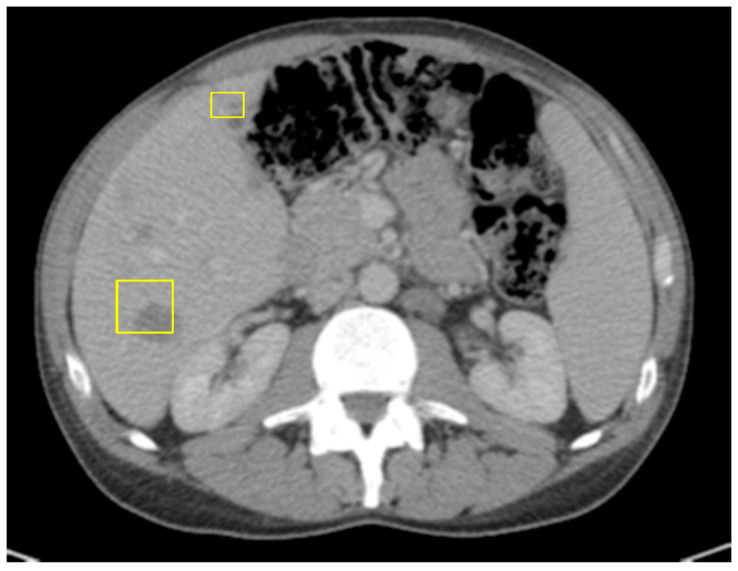
CT of abdomen/pelvis right after Pembrolizumumab: slight decrease in size of hepatic masses, stable retroperitoneal lymph nodes, and small splenomegaly.

**Figure 6 curroncol-32-00370-f006:**
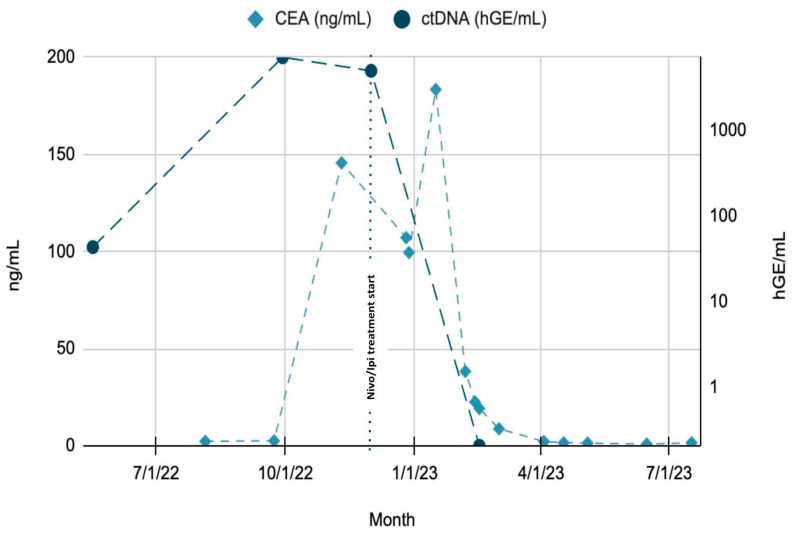
Trends of CEA and ctDNA throughout patient’s clinical course. Green region: rising ctDNA levels prior to treatment initiation. Blue region: rapid decline in ctDNA from Month 8 to Month 10, following initiation of Nivolumab + Ipilimumab. CEA lag: decline in CEA noted after ctDNA drop, suggesting delayed response by conventional marker. Timeline annotations: key therapeutic events, such as treatment initiation (Month 8) and any subsequent modifications or responses, are clearly marked to aid interpretation.

**Figure 7 curroncol-32-00370-f007:**
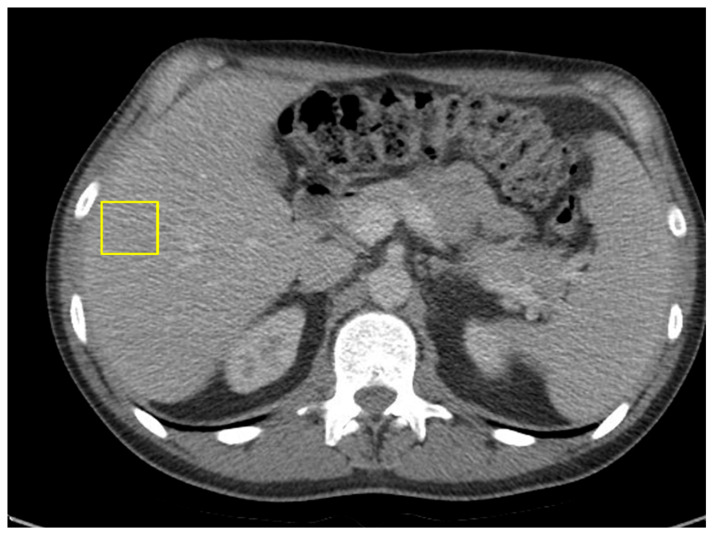
CT of abdomen and pelvis showing resolution of hepatic masses, with no new lymphadenopathy.

## Data Availability

All data generated or analyzed during this study are included in this article. Further enquiries can be directed to the corresponding author.

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
