# Peer review of "Unveiling ctDNA Response: Immune Checkpoint Blockade Therapy in a Patient with POLE Mutation-Associated Early-Onset Colon Cancer"

_curroncol, 2025, doi:10.3390/curroncol32070370_

Round 1
Reviewer 1 Report
Comments and Suggestions for Authors
In this case report, Ramachandran et al report on a case of colorectal cancer in a 45-y.o. male which progressed to hepatic metastases following neoadjuvant chemotherapy and resection of primary. Progression continued on chemotherapy + angiogenesis inhibitor and may have stabilized on pembrolizumab (PD-1 inhibitor). A switch to dual nivolumab (PD-L1 inhibitor)/ipilimumab CTLA4 inhibitor) therapy was associated with increased tumor burden on imaging as well as CEA levels; however, ctDNA levels decreased. After switching to single-agent nivolumab due to poor tolerance of ipilimumab, CEA decreased, and after 2 years on-treatment CT imaging shows no signs of residual disease. This represents an interesting example of how some off-label alternative IO approaches may be effective – although this cannot be predicted up front.
Suggestions:
- Changes in on-treatment ctDNA levels appear to precede those of CEA (Graph 1) – may be worth discussing this, since it may act as an earlier indicator of future treatment response based on the overall disease course reported.
- Also for Graph 1, adding indications along the timeline re: therapy start/change would make it easier to interpret.
- Minor point – using dashed vs. solid lines for the ctDNA vs. CEA plots would make it easier to distinguish them. Might also be simpler to add months as numbers on X-axis instead of inside the graph.
- Based on the title, ctDNA assay was an important element in patient monitoring – perhaps additional detail regarding the methodology used would be of interest. Signatera panel is personalized – which mutations were tracked here? POLE, BRAF and PIK3CA?
- Unclear as to whether anti-PD-1 or anti-PD-L1 immunohistochemistry was performed on the clinical specimen from primary tumor - if so, information regarding results would help to improve understanding of this case. Since MMR proteins and POLE were assessed by IHC, likely that PD-1/PD-L1 IHCs were also performed? It appears that hepatic metastases were not biopsied, so information similar regarding these would be lacking.
- Information re: point 5 above may indicate why nivolumab appears to have been more effective than pembrolizumab ...
- Some minor grammatical errors, e.g., line 47, should be "and MMR"; line 54, stray comma before "(PD-1)"; line 55, "Cytotoxic" does not require capitalization; line 65, "who presented", etc.
Please see point 7 above.
Author Response
Aaron J Scott MD
Associate Professor of Medicine
Director, Early Therapeutics Program
University of Arizona Cancer Center
Date: 6/18/2025
RE: curroncol-3659185 submission
Dear Editorial Board:
Thank you for your interest in our case report and thank you to the reviewers for their helpful comments and edit suggestions. We believe we have incorporated all suggestions into this latest draft of the case report. Please find our responses to these edits/suggestions below.
Reviewer question 1- Changes in on-treatment ctDNA levels appear to precede those of CEA (Graph 1) – may be worth discussing this, since it may act as an earlier indicator of future treatment response based on the overall disease course reported.
Reviewer question 2- Also for Graph 1, adding indications along the timeline re: therapy start/change would make it easier to interpret.
Author response to 1 and 2 - Thank you for the valuable suggestions. We have revised the paragraph to clearly describe the temporal sequence of changes in circulating tumor DNA (ctDNA) and carcinoembryonic antigen (CEA), particularly in relation to the initiation of nivolumab and ipilimumab therapy at Month 8. The updated text highlights how ctDNA decline preceded the drop in CEA and aligns with clinical improvement. We have also updated Graph 1 to include annotations for key treatment milestones and color-coded regions to visually distinguish these trends. We believe this revision enhances clarity and better illustrates the potential of ctDNA as an early biomarker of treatment response.
Reviewer question 3- Minor point – using dashed vs. solid lines for the ctDNA vs. CEA plots would make it easier to distinguish them. Might also be simpler to add months as numbers on X-axis instead of inside the graph.
Author response to 3 - Thank you for your valuable suggestion. We have updated the figure accordingly by using dashed and solid lines to better distinguish between ctDNA and CEA plots. Additionally, we have modified the X-axis to display months as numerical values, which simplifies interpretation. We believe these changes improve the clarity and readability of the figure.
Reviewer question 4- Based on the title, ctDNA assay was an important element in patient monitoring – perhaps additional detail regarding the methodology used would be of interest. Signatera panel is personalized – which mutations were tracked here? POLE, BRAF and PIK3CA?
Author response to 4 - Thank you for your insightful comment. We have revised the case report to provide additional detail regarding the ctDNA assay methodology. Specifically, we now clarify that Signatera™ was used as the personalized, tumor-informed ctDNA assay, developed using whole-exome sequencing of the patient’s tumor and matched normal tissue to identify up to 16 clonal somatic variants for longitudinal tracking. In this case, the custom Signatera panel incorporated tumor-specific mutations including POLE, BRAF (p.D594G), and PIK3CA (p.R88Q). We also emphasized how ctDNA dynamics preceded changes in conventional markers like CEA and corresponded with the patient’s clinical course, supporting its utility in monitoring treatment response.
Reviewer question 5- Unclear as to whether anti-PD-1 or anti-PD-L1 immunohistochemistry was performed on the clinical specimen from primary tumor - if so, information regarding results would help to improve understanding of this case. Since MMR proteins and POLE were assessed by IHC, likely that PD-1/PD-L1 IHCs were also performed? It appears that hepatic metastases were not biopsied, so information similar regarding these would be lacking.
Author Response to 5: Thank you for this thoughtful and important comment. We appreciate the opportunity to clarify. In this case, PD-L1 immunohistochemistry (IHC) was performed on the primary tumor as part of extended immuno-oncology profiling. The tumor proportion score (TPS) for PD-L1 was <1%, and immune cell staining was also low. Despite low PD-L1 expression, the patient demonstrated a robust and durable clinical response to immune checkpoint inhibition, likely driven by the ultramutated POLE mutation and high TMB (87.9 m/MB), both of which have been associated with increased tumor immunogenicity and responsiveness to immunotherapy regardless of PD-L1 status. As noted by the reviewer, the hepatic metastases were not biopsied, and therefore PD-L1 IHC data specific to the metastatic lesions is not available.
Reviewer question 6- Case reports should include a succinct introduction about the general medical condition or relevant symptoms that will be discussed in the case report; the case presentation including all of the relevant de-identified demographic and descriptive information about the patient(s), and a description of the symptoms, diagnosis, treatment, and outcome;
Author response to 6- Thank you for your guidance. We have revised the case report to include a concise introduction outlining the general medical condition and relevant symptoms discussed. The case presentation now provides all pertinent de-identified demographic and descriptive information about the patient, along with a detailed description of the symptoms, diagnosis, treatment, and outcome, in accordance with the recommended structure.
Reviewer question 7- Please amend ”…POLE-Mutation Associated Colon Cancer Patient” to “…Patient with POLE-Mutation Associated Early Onset Colon Cancer”
Author response to 6- Thank you for your suggestion. We have amended the title as requested to “…Patient with POLE-Mutation Associated Early Onset Colon Cancer” to better reflect the case description.
Reviewer question 8- Finally, please add a conclusion to briefly outline the take-home message and the lessons learned as per the Author guidelines, in order to further highlight the importance of the obtained results in the context of modern immunotherapy in oncology
Author response to 8- Thank you for your valuable recommendation. We have added a conclusion to the manuscript that summarizes the key take-home message and lessons learned from this case. The conclusion emphasizes the significance of our findings in the context of modern immunotherapy in oncology, highlighting the potential implications of POLE-mutation associated early onset colon cancer for personalized treatment approaches.
Reviewer clarifications - Some minor grammatical errors, e.g., line 47, should be "and MMR"; line 54, stray comma before "(PD-1)"; line 55, "Cytotoxic" does not require capitalization; line 65, "who presented", etc.
Author response -Thank you for your careful review and attention to detail. We have corrected the noted grammatical errors throughout the manuscript, including:
- Revised line 47 to read “and MMR” for clarity and accuracy.
- Removed the stray comma before “(PD-1)” on line 54.
- Corrected the capitalization of “cytotoxic” on line 55.
- Amended the phrasing on line 65 to “who presented” for improved grammar.
Sincerely,
Aaron J Scott MD
Associate Professor of Medicine
Director, Early Therapeutics Program
University of Arizona Cancer Center
Reviewer 2 Report
Comments and Suggestions for Authors
The case study addresses an up-to-date translational question in oncology. Analyses were performed using appropriate methods and study limitations are well documented. The manuscript might be accepted after careful review.
Major comments
Title & text
Please avoid the phrase “cancer patient” throught the text and title, and amend to “patient with cancer”
Please amend ”…POLE-Mutation Associated Colon Cancer Patient” to “…Patient with POLE-Mutation Associated Early Onset Colon Cancer”
Abstract
The abstract is written as a journal club on the topic, not depicting the obtained data. Please rewrite the abstract so it includes a valid presentation of the case as per the Author guidelines:
“Case reports should include a succinct introduction about the general medical condition or relevant symptoms that will be discussed in the case report; the case presentation including all of the relevant de-identified demographic and descriptive information about the patient(s), and a description of the symptoms, diagnosis, treatment, and outcome; a discussion providing context and any necessary explanation of specific treatment decisions; a conclusion briefly outlining the take-home message and the lessons learned.
Minor comments
- Lines 59-61 please amend to “Our paper aims to showcase the effectiveness of nivolumab and ipilimumab in patients with POLE mutant CRC”
- Lines 91-93 please add information on the method used for molecular profiling
- Line 116 please amend to ctDNA
Finally, please add a conclusion to briefly outline the take-home message and the lessons learned as per the Author guidelines, in order to further highlight the importance of the obtained results in the context of modern immunotherapy in oncology. A short mention on the economic burden and feasibility of the introduction of ctDNA measurement in everyday clinical practice should also be inserted.
Author Response

(The authors gave the same response as above.)
